# Sustained North Atlantic warming drove anomalously intense MIS 11c interglacial

Hsun-Ming Hu [1,2,3] ✉, Gianluca Marino [4] ✉, Carlos Pérez-Mejías[5], Christoph Spötl [6], Yusuke Yokoyama [7,8], Jimin Yu [9,10], Eelco Rohling [11,12], Akihiro Kano[7], Patrick Ludwig [13], Joaquim G. Pinto [13], Véronique Michel[14,15], Patricia Valensi[16], Xin Zhang[17], Xiuyang Jiang[17], Horng-Sheng Mii [18], Wei-Yi Chien[1], Hsien-Chen Tsai[1], Wen-Hui Sung[1], Chia-Hao Hsu[1], Elisabetta Starnini[19], Marta Zunino[20] & Chuan-Chou Shen [1,3] ✉

The Marine Isotope Stage (MIS) 11c interglacial and its preceding glacial termination represent an enigmatically intense climate response to relatively weak insolation forcing. So far, a lack of radiometric age control has confounded a detailed assessment of the insolation-climate relationship during this period. Here, we present $^{230}$Th-dated speleothem proxy data from northern Italy and compare them with palaeoclimate records from the North Atlantic region. We find that interglacial conditions started in subtropical to middle latitudes at 423.1 ± 1.3 thousand years (kyr) before present, during a first weak insolation maximum, whereas northern high latitudes remained glaciated (sea level ~ 40 m below present). Some 14.5 ± 2.8 kyr after this early subtropical onset, peak interglacial conditions were reached globally, with sea level 6–13 m above present, despite weak insolation forcing. We attribute this remarkably intense climate response to an exceptionally long (~15 kyr) episode of intense poleward heat flux transport prior to the MIS 11c optimum.

The fifth last transition from glacial to interglacial conditions (Termination V, T-V) at ~428 thousand years before present (kyr BP) (ref. 1), and the ensuing interglacial period known as Marine Isotope Stage (MIS) 11c (Fig. 1), seem to challenge the orbital theory of Pleistocene glacial-interglacial cycles in a fundamental manner[2–6]. Arguably the largest amplitude termination[4] and the longest[7] (~30 kyr) and second warmest[1,7–9], interglacial with highest global sea level[10–13] coincided with weak insolation forcing[14,15] (Fig. 1); this is known as the "MIS 11

[1]High-Precision Mass Spectrometry and Environment Change Laboratory (HISPEC), Department of Geosciences, National Taiwan University, Taipei 10617 ROC, Taiwan. [2]Radiogenic Isotope Facility, School of Earth and Environmental Sciences, The University of Queensland, Brisbane QLD 4072, Australia. [3]Research Center for Future Earth, National Taiwan University, Taipei 10617 ROC, Taiwan. [4]Centro de Investigación Mariña, GEOMA, Palaeoclimatology Lab, Universidade de Vigo, Vigo 3610, Spain. [5]Institute of Global Environmental Change, Xi'an Jiaotong University, 710049 Xi'an, China. [6]Institute of Geology, University of Innsbruck, Innrain 52, 6020 Innsbruck, Austria. [7]Atmosphere and Ocean Research Institute, The University of Tokyo, 5-1-5 Kashiwanoha, Kashiwa, Chiba 277-8564, Japan. [8]Department of Earth and Planetary Science, Graduate School of Science, The University of Tokyo, 7-3-1 Hongo, Bunkyo-ku, Tokyo 113-0033, Japan. [9]Laoshan Laboratory, Qingdao 266237, China. [10]Research School of Earth Sciences, The Australian National University, Canberra ACT 2601, Australia. [11]Department of Earth Sciences, Utrecht University, 3584 CB Utrecht, Netherlands. [12]Ocean and Earth Science, University of Southampton, National Oceanography Centre, Southampton SO14 3ZH, UK. [13]Institute of Meteorology and Climate Research Troposphere Research (IMKTRO), Karlsruhe Institute of Technology (KIT), Karlsruhe, Germany. [14]Université Côte d'Azur, CNRS, CEPAM, 06300 Nice, France. [15]Université Côte d'Azur, CNRS, OCA, IRD, Géoazur, 06560 Valbonne, France. [16]UMR7194 HNHP (MNHN-CNRS-UPVD), Institut de Paléontologie Humaine, 75013 Paris, France. [17]Key Laboratory of Humid Subtropical Eco-Geographical Processes, Ministry of Education, College of Geography Science, Fujian Normal University, Fuzhou 350007, China. [18]Department of Earth Sciences, National Taiwan Normal University, Taipei 11677 ROC, Taiwan. [19]Department of Civilizations and Forms of Knowledge, University of Pisa, Via dei Mille 19, 56126 Pisa, Italy. [20]Toirano Cave, Piazzale D. Maineri 1, 17055 Toirano (SV), Italy. ✉e-mail: hsunming.hu@gmail.com; gianluca.marino@uvigo.es; river@ntu.edu.tw

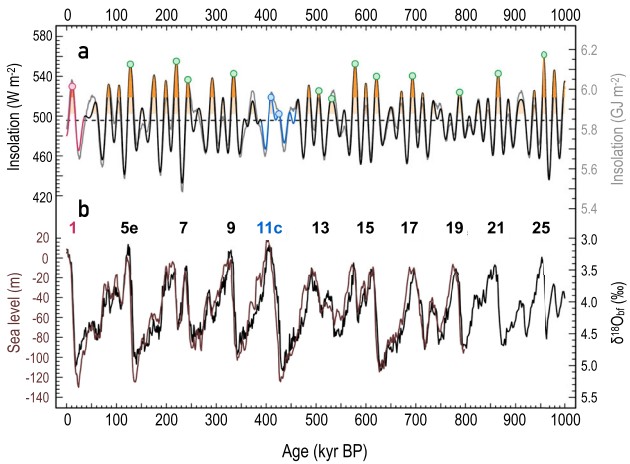

**Fig. 1 | Glacial-interglacial cycles and insolation changes of the last million years. a** June 21st insolation (black)[14] and caloric summer half-year insolation energy (grey)[5] at 65°N. Intervals of Marine Isotope Stage (MIS) 11c (smalt blue) and 1 (the Holocene, regal red) are highlighted. Solid dots indicate insolation peaks associated with the interglacial periods, including MIS 11c (smalt blue) and MIS 1 (regal red). The dashed black line is the mean insolation (496.3 W m$^{-2}$; 5.845 GJ m$^{-2}$) of the last million years (cf. Mitsui et al., ref. 20), while the lighter and darker orange shading indicates insolation values higher than the first (502.3 W m$^{-2}$) and second (518.8 W m$^{-2}$) insolation peaks of MIS 11c, respectively. **b** Black: stacked benthic foraminiferal stable oxygen isotope ($\delta^{18}O_{bf}$) record[1], which reflects changes in global ice volume and temperature. Brown: a global sea-level reconstruction[10]. Odd numbers denote interglacial MIS of the last million years.

paradox". It has been suggested that the duration and magnitude of MIS 11c warmth, particularly in the North Atlantic[16,17], may have driven extensive Greenland ice sheet mass loss[18,19], contributing to sea-level rise up to 6–13 m above the present level[11–13]. MIS 11c encompasses two boreal summer insolation peaks[14]. The first, weaker peak, is centred on 425.6 kyr BP, and the second on 409.5 kyr BP (Fig. 1a). Recently, Mitsui et al.[20] suggested that terminations of the last million years generally started and ended when a 5.845 GJ m$^{-2}$ threshold of average caloric summer half-year insolation was crossed (Fig. 1a). Accordingly, the second rather than first insolation peak of MIS 11c should have led to the onset of interglacial conditions, but this is at odds with evidence of sustained warm[17,21] and/or humid[22–25] conditions in the wider North Atlantic during the first half of MIS 11c. However, the chronology of these North Atlantic records is either based on orbital tuning, on correlation with far-field records (Antarctic ice cores), or on a few datable tephra layers. This precludes a robust evaluation of the timing of climate change relative to insolation prior to and during MIS 11c (ref. 7) in this critical region for the interaction between ice sheets, ocean circulation, and atmosphere[26].

Here we present a precisely dated record of climate changes leading to MIS 11c from Bàsura cave in northern Italy. The cave is well-sited to provide insight into subtropical/mid-latitude climate changes because the western Mediterranean Sea is sensitive to the North Atlantic Ocean and climate change, given that the dominant westerly winds efficiently transport heat and moisture towards the basin[27–29], while its restricted, land-locked nature with limited inertia tends to amplify climate signals. Previous studies exploited this regional sensitivity to provide important insights into the timing and nature of other glacial-interglacial transitions[30–36]. Our radiometrically dated speleothem multi-proxy records for the 480–360 kyr interval span glacial MIS 12, T-V, and MIS 11c. We discuss these in a context of palaeoclimate evidence from the wider North Atlantic and use the protracted nature of T-V to obtain a detailed view of the sequence of processes during the deglaciation.

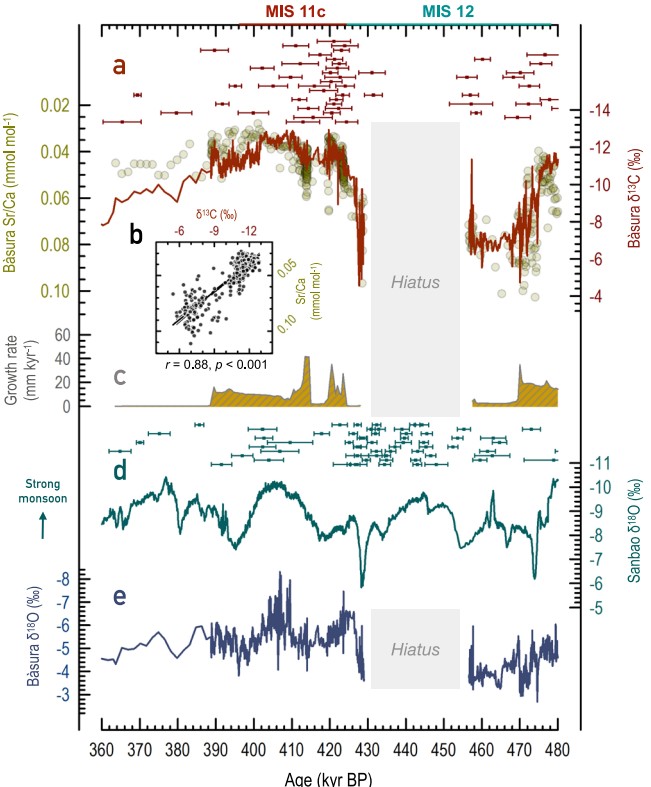

**Fig. 2 | Speleothem time series between 480 and 360 thousand years before present (kyr BP). a** Strontium to calcium ratios (Sr/Ca, circles) and stable carbon isotopes ($\delta^{13}C$, solid line) measured in flowstone BA7-1 from Bàsura cave (northern Italy, this study). **b** Cross plot ($r = 0.88$, $p < 0.001$) of $\delta^{13}C$ and Sr/Ca data in BA7-1 (this study). **c** BA7-1 growth rate (this study). **d** Speleothem $\delta^{18}O$ from Sanbao cave in eastern China[48]. **e** BA7-1 stable oxygen isotopes ($\delta^{18}O$, this study). The 429–457 kyr BP hiatus is indicated by the grey shading. Marine Isotope Stages (MIS) 11c and 12 defined in ref. 1 are given at the top. Color-coded error bars in (**a**) and (**d**) are the U-Th ages with 2σ errors from Bàsura and Sanbao caves, respectively.

## Results and discussion

Bàsura cave (44°08'N, 8°12'E, 200 m above sea level; Supplementary Fig. 1) is located in the coastal region of the Gulf of Genoa, which features a typical Mediterranean climate with mild and humid winters and hot and dry summers[37–39]. Centennial-resolution time series of stable carbon ($\delta^{13}C$) and oxygen ($\delta^{18}O$) isotopes (Methods; Supplementary Data 1) and strontium to calcium (Sr/Ca) ratios (Methods; Supplementary Data 2) were generated along flowstone BA7-1 (Supplementary Fig. 2). Hendy tests[40] (Supplementary Fig. 3) support previous evidence that calcite precipitation occurs close to isotopic equilibrium in Bàsura cave[37,39]. The chronology of BA7-1 hinges on 54 high-precision U-Th ages (Methods; Supplementary Data 3), processed using both StalAge algorithm[41] and OxCal software[42] (Supplementary Fig. 4). Given the statistically identical outputs, in the following we use the chronology from StalAge algorithm[41].

A central feature of the BA7-1 records is strong covariation between the $\delta^{13}C$ and Sr/Ca time series ($r = 0.88$, $p < 0.001$, 480–390 kyr BP; Fig. 2a and b), with both generally mirroring changes in flowstone growth rates, i.e., low $\delta^{13}C$ and Sr/Ca match high growth rates (Fig. 2c). A large, positive shift in both $\delta^{13}C$ and Sr/Ca at 476–468 kyr BP coincides with an abrupt reduction in flowstone growth rates and leads to long-term $\delta^{13}C$ and Sr/Ca maxima (growth rate minimum) until 457 kyr BP. A hiatus between 457 and 429 kyr BP is followed by major $\delta^{13}C$ and Sr/Ca decreases centred at 427 kyr BP, accompanied by increasing growth rates. This transition leads to two distinct and broad $\delta^{13}C$ and Sr/Ca minima at 424–418 kyr BP and 412–402 kyr BP, separated by a 6-kyr-long interlude of higher values.

Speleothem $\delta^{13}C$ is sensitive to both temperature and rainfall[37,43] through their impacts on vegetation, soil bioactivity, and the extent of prior carbonate precipitation (PCP)[44] (Methods). High $\delta^{13}C$ indicates cold and/or dry climatic conditions with low soil bioactivity or enhanced PCP, with opposite conditions indicated by low $\delta^{13}C$. Sr/Ca is also tied to the PCP extent at Bàsura cave[38]. High Sr/Ca reflects dry conditions with long karst water residence times and enhanced PCP, and vice versa for low Sr/Ca. Covariation between $\delta^{13}C$ and Sr/Ca in BA7-1 (Fig. 2b), along with a remarkable similarity between our $\delta^{13}C$ and a precipitation proxy from northern Greece[45] (Supplementary Fig. 5), collectively argue for a strong Bàsura cave $\delta^{13}C$ sensitivity to precipitation amount changes.

High $\delta^{13}C$ and slow or ceased flowstone growth between 468 and 428 kyr BP in BA7-1 (Fig. 2a and c) attest to arid conditions at Bàsura cave during glacial MIS 12, in agreement with regional pollen data[22–25]. At $427 \pm 2$ kyr BP, there is a rapid transition toward more humid conditions typical of the regional interglacial climate[45–47], although assessment of the full amplitude of the transition is precluded by the hiatus. Timing of the transition agrees within uncertainties ($\pm 1$–2 kyr, $2\sigma$) with that of a T-V weak monsoon interval in Sanbao cave[48] (China, Fig. 2d), which to date provided the only radiometric chronological constraints on T-V. A contemporaneous negative ~3 ‰ shift in BA7-1 $\delta^{18}O$ (Fig. 2e) further supports the agreement between the Bàsura and Sanbao cave records on the timing of T-V, given that it plausibly arises from a combination of amount[39,49] and source-water effects[31–33,39,49] (Methods). The latter relates to the discharge of isotopically light meltwater from waning ice sheets into the North Atlantic and from Alpine deglaciation into the Gulf of Genoa via the Rhône River, followed by transfer of the more negative meltwater $\delta^{18}O$ signatures via evaporation and subsequent rainfall to the Bàsura cave site.

## Radiometric age constraints of MIS 11c

The two broad $\delta^{13}C$ and Sr/Ca minima in BA7-1 indicate two periods of peak interglacial conditions, which occurred at 424–418 kyr BP and 412–402 kyr BP. Their onsets coincided within uncertainties ($\pm 2$ to 3 kyr, $2\sigma$) with the MIS 11c boreal summer insolation peaks[14] at 425.6 and 409.5 kyr BP (Fig. 3a). Probabilistic analysis (Methods) of the $\delta^{13}C$ record and its associated uncertainties highlights two interglacial maxima of similar magnitude at Bàsura cave (Fig. 3a). Two maxima of approximately similar magnitude are also observed in other marine and terrestrial records of MIS 11c from the North Atlantic[17,21–25]. The Bàsura chronology was, thus, transferred to North Atlantic temperature records (Methods) to provide radiometric constraints to the MIS 11c regional warming. This portrays the nature of the climate responses to the relatively weak insolation changes during MIS 11c[14,15] in this critical region for glacial-interglacial climate change[26], on an absolutely constrained, radiometric timescale.

Today, rainfall in the Bàsura cave area occurs primarily in autumn and winter, and is largely governed by westerly winds and storms[27–29,39,50] that deliver relatively warm, moisture-laden air masses from the mid-latitude northeastern Atlantic Ocean. Precipitation amount and seasonality at Bàsura cave is therefore strongly related to latent heat flux in the moisture-source sector of the Atlantic Ocean[27] (Supplementary Fig. 6), which explains the correlation between Bàsura cave precipitation amount and sea surface temperature (SST) (Methods). This connection holds on glacial-interglacial timescales[51–54] because North Atlantic warming along with the steep land-sea temperature gradient tied to winter cooling of the Mediterranean borderlands enhance the influence of storm activity (from both Atlantic origin and Mediterranean cyclogenesis) over the basin[47,54–56]. These notions underpin synchronisation of the alkenone-based SST records from mid-latitude North Atlantic Integrated Ocean Drilling Program (IODP) Site U1313 (41°00′N, 32°57′W, 3426 m water depth; ref. 21; Supplementary Data 4) and core MD03-2699 (39°02′N, 10°40′W, 1865 m water depth; ref. 57; Supplementary Data 4) to BA7-1 $\delta^{13}C$

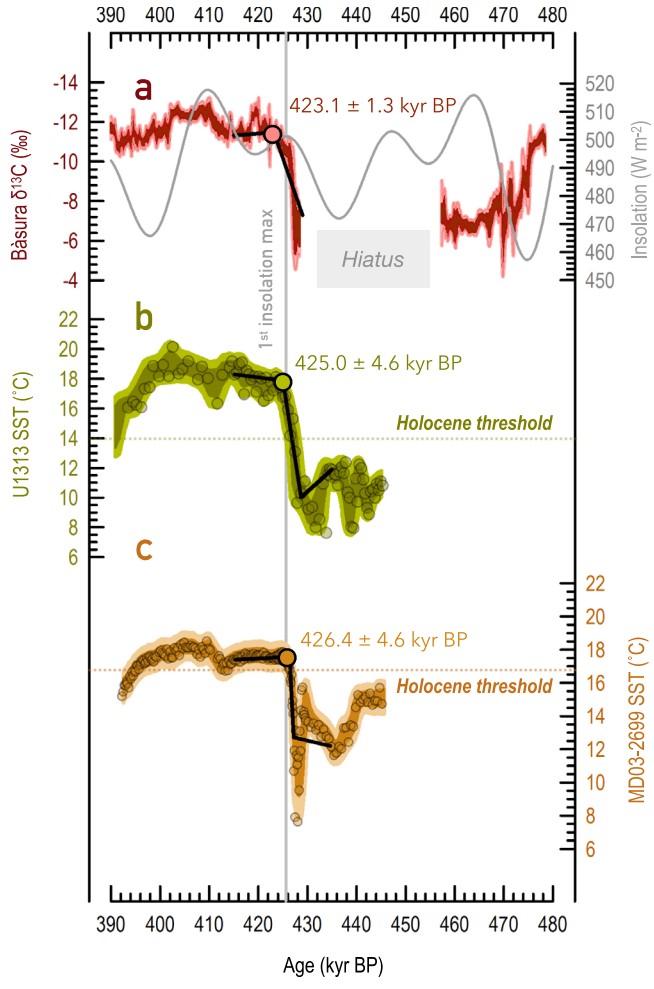

**Fig. 3 | Radiometrically constrained time series of climate change during glacial termination V (T-V) and Marine Isotope Stage (MIS) 11c. a** Red: Bàsura $\delta^{13}C$, whereby dark and light red shaded envelopes indicate the 68 and 95% confidence limits, respectively. Grey: June 21st insolation at 65°N (ref. 14). Vertical grey line indicates the first insolation maximum in early MIS 11c at 425.6 thousand years ago (kyr BP). **b** Alkenone-based sea-surface temperature (SST) from Site U1313 (ref. 21) on its radiometrically constrained chronology (this study). Dark and light olive-shaded envelopes indicate the 68 and 95% confidence limits, respectively. **c** Alkenone-based SST record from core MD03-2699 (ref. 57) on its radiometrically constrained chronology (this study). Dark and light bronze-shaded envelopes indicate the 68 and 95% confidence limits, respectively. In (**b**) and (**c**), light-coloured dashed horizontal lines indicate the Holocene temperature threshold values. In **a**–**c**, the black lines and color-coded circles represent the results of the BREAKFIT/RAMPFIT change-point fitting analysis[59,60], with a search window of 429–415 (**a**) and 435–415 kyr BP (**b** and **c**). The corresponding ages and errors ($2\sigma$) are denoted.

(Methods; Supplementary Fig. 7). This synchronisation is supported by the agreement between the timing of low SSTs during the Heinrich-like stadial of T-V in Site U1313 and MD03-2699 (Supplementary Fig. 8a, b, and d), and that of the weak monsoon interval in Sanbao cave[48] (Supplementary Fig. 8c). Weak monsoon intervals in Sanbao cave have been used to constrain the timing of glacial terminations based on monsoon sensitivity to North Atlantic meltwater pulses and Heinrich stadial conditions[48,58].

Our assessment of the onset of the MIS 11c interglacial in the subtropical North Atlantic hinges on two independent approaches. First, we employed change-point analysis using the RAMPFIT algorithm[59] to statistically determine when the SST rise associated with T-V "breaks" into the plateau that denotes the onset of interglacial conditions. Based on our radiometrically constrained age model

**Table 1 | Timing of the onset of MIS 11c**

| Method | Core | Tuned age (kyr BP) | Error (kyr, 2σ) | Original age (kyr BP) | Error (kyr, 2σ) |
|---|---|---|---|---|---|
| RAMPFIT analysis[59] | U1313 | 425.0 | ± 4.6 | 424.3 | ± 4.0 |
| | MD03-2699 | 426.4 | ± 4.6 | 426.2 | ± 4.0 |
| Holocene threshold | U1313 | 425.5 | ± 4.4 | 426.1 | ± 4.0 |
| | MD03-2699 | 425.1 | ± 4.4 | 425.6 | ± 4.0 |

(Methods), the RAMPFIT analysis that passed sensitivity tests (Methods; Supplementary Table 1) indicates that MIS 11c initiated at 425.0 ± 4.6 kyr BP (2σ) at Site U1313 (Fig. 3b) and at 426.4 ± 4.6 kyr BP (2σ) in core MD03-2699 (Fig. 3c). Application of BREAKFIT[60] to the Bàsura cave $\delta^{13}$C yields an age of 423.1 ± 1.3 kyr BP (2σ) for the onset of MIS 11c at this location (Fig. 3a).

Second, we applied a threshold approach to determine the onset of MIS 11c at Site U1313 and in core MD03-2699 in an alternative, independent manner. This defines interglacial conditions based on exceedance of minimum Holocene SST recorded at the same location. The Holocene is a well-suited reference for evaluating the MIS 11c warming anomaly because it occurred under a similar (low-eccentricity) orbital configuration[14,15], while its temporal evolution and spatial patterns[61] are well understood based on a wealth of well-dated palaeoclimate records[62]. On this basis, MIS 11c would span the interval at Site U1313 (ref. 21) and MD03-2699 (ref. 57) through which the lower bound of the 95% SST confidence level equals or exceeds the minimum Holocene SST (hereafter Holocene threshold) at the same locations[21,57] (14.0°C for Site U1313 [ref. 21], 16.7°C for MD03-2699 [refs. 63]). Given our radiometrically constrained age models (Methods), the Holocene threshold was surpassed at 425.5 ± 4.4 kyr BP at Site U1313 (Fig. 3b) and at 425.1 ± 4.4 kyr BP (2σ) in core MD03-2699 (Fig. 3c). These estimates agree within uncertainties with those from the change-point analysis reported above.

We also tested the RAMPFIT[59] and Holocene threshold methods to determine the MIS 11c onset using the original published chronologies for Site U1313 (ref. 21) and MD03-2699 (ref. 57). On those chronologies, change-point analysis suggests that MIS 11c commenced at 424.3 ± 4.0 kyr BP (Site U1313 [ref. 21]) and 426.2 ± 4.0 kyr BP (Site MD03-2699 [ref. 57]), while the Holocene threshold approach yields 426.1 ± 4.0 kyr BP and 425.6 ± 4.0 kyr BP, respectively. Taken together, all the suggested MIS 11c time points (Table 1) obtained by two detection approaches on various archives, and using either our synchronized ages or the original age models, give a consistent result that agrees within uncertainties with the breaking-point age[60] estimated from Bàsura $\delta^{13}$C at 423.1 ± 1.3 kyr BP (2σ). We infer that North Atlantic warming occurred at the same time (within uncertainties) as Mediterranean hydroclimate intensification. The onset of interglacial conditions early in MIS 11c was not limited to subtropical and middle latitudes in the North Atlantic[64]. The early climatic optimum appears to have extended as far north as southern Greenland[17] and southern Iceland[65], likely due to sustained northward heat transport from the subtropical gyre by the North Atlantic Current[66]. These results show that the onset of MIS 11c in the wider North Atlantic/Mediterranean coincided (within uncertainties) with the first, subtle 65°N summer insolation maximum (ref. 14) of 425.6 kyr BP (Fig. 4a).

### Climate versus ice volume during MIS 11c

Our radiometric chronology indicates that from 423.1 ± 1.3 kyr BP, humid and warm interglacial climate conditions prevailed in the North Atlantic Ocean and Mediterranean region (Fig. 4a and b), while the Northern Hemisphere ice sheets still accounted for a sea level at 41 ± 16 m (2σ) below present, as documented by the radiometrically constrained Red Sea sea-level reconstructions[67] (Fig. 4c). This continued presence of substantial continental ice volumes implies that full interglacial climate conditions had yet to be reached globally[7,68]. Sea

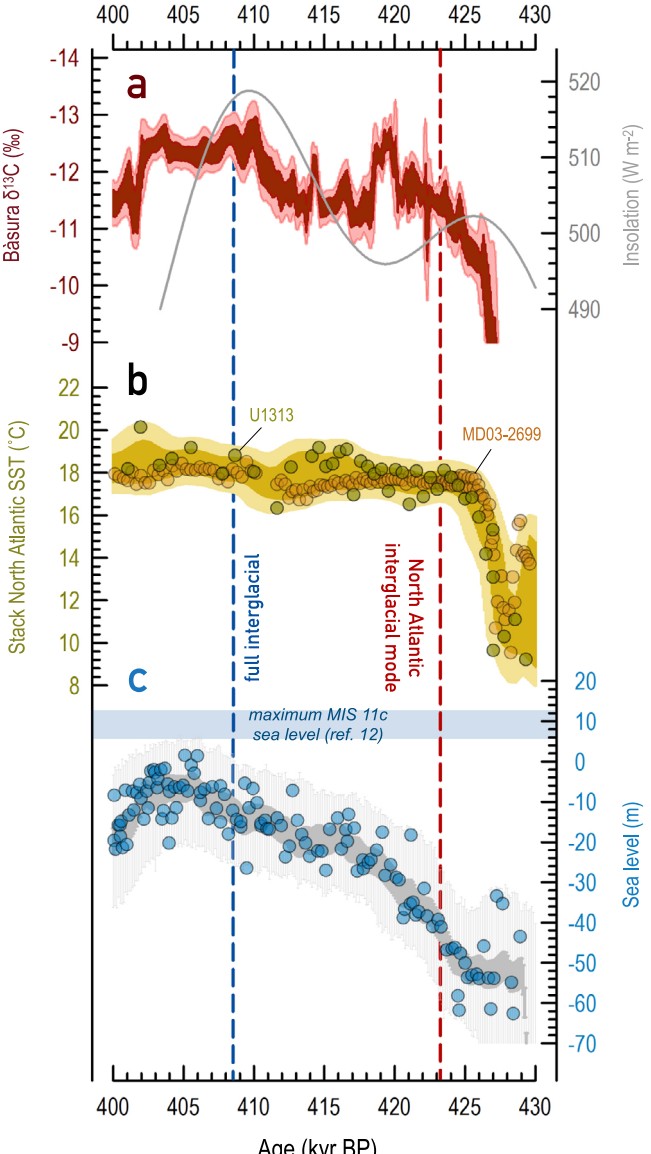

**Fig. 4 | Radiometrically constrained palaeoclimate records of the Marine Isotope Stage (MIS) 11c interglacial. a** Red: Bàsura $\delta^{13}$C, whereby dark and light red shaded envelopes indicate the 68 and 95% confidence limits, respectively. Grey: June 21st insolation at 65°N (ref. 14) **b** Stacked alkenone-based sea-surface temperature (SST) record from cores Site U1313 (ref. 21) and MD03-2699 (ref. 57) on a radiometrically constrained chronology (this study). Olive and bronze circles indicate the datapoints of cores Site U1313 and MD03-2699. Darker and lighter shaded envelopes indicate the 68 and 95% confidence limits, respectively. **c** Radiometrically constrained relative sea-level reconstruction from Red Sea[67]. Blue circles indicate the original datapoints. Darker and lighter grey shaded envelopes indicate the 68 and 95% probability envelopes, respectively. Blue shade denotes the estimate of maximum MIS 11c sea level[12]. Dark red and blue vertical dashed lines mark the onset of the Atlantic interglacial mode and global interglacial status, respectively.

level reached today's levels at 408.6 ± 2.5 kyr BP (ref. [67]), and 6–13 m above current levels[11] thereafter (Fig. 4c). Our analysis suggests that ice sheet reduction that drove sea level to positions equal to, or higher than, common interglacial levels was achieved only 14.5 ± 2.8 kyr after the emergence of warm and humid conditions in the subtropical and mid-latitude North Atlantic. Hence, the first weak boreal summer insolation peak at 425.6 kyr BP (Fig. 4a) could have caused a transition to interglacial warmth and hydroclimate in the North Atlantic and the Mediterranean regions (Fig. 4a and b), but was not sufficient for completing the transition to full interglacial conditions globally. The latter only occurred some ~15 kyr later, with a major ice-volume reduction (Fig. 4c) exemplified by wide-spread ice-free conditions in Greenland[19].

During late glacial MIS 12, peak cold and dry conditions developed in the Mediterranean region[25,45,46], which is consistent with the interruption of speleothem growth (hiatus) of BA7-1 (Fig. 2c). North Atlantic ice sheets reached their maximum extent of the last several glacial-interglacial cycles[69]. This large extent made the southern margins of the Northern Hemisphere ice sheets particularly sensitive to even small increases in boreal insolation[70], like that from 436.4 to 425.6 kyr BP (Fig. 1a). This caused decay of the southernmost, most vulnerable margins of the Northern Hemisphere ice sheets, driving ~80 m of sea-level rise[10,67] (considering a sea-level minimum at approximately −120 m in late MIS 12 [refs. 10,67]; Fig. 1b).

The emergence of warm and humid conditions in the subtropical and mid-latitude North Atlantic from 423.1 ± 1.3 kyr BP coincided with the onset of an interglacial mode of the Atlantic Meridional Overturning Circulation (AMOC), whereby North Atlantic Deep Water ventilated the deep Atlantic Ocean[71]. A coupled general circulation model[72] simulates a vigorous AMOC during MIS 11c, leading to anomalously strong northward heat transport from the sub-tropical latitudes. This picture agrees with warming documented early in MIS 11c in the eastern North Atlantic[66], further north at Eirik Drift[17], and in southern Greenland[73]. A strong AMOC[72] since early MIS 11c sustained protracted (~15 kyr) warming in northern high latitudes, which has been proposed to be key to the extensive Greenland ice sheet reduction that sets MIS 11c apart from other middle to late Pleistocene interglacial periods[16]. These climate developments preconditioned the Earth system for reaching the intense MIS 11c interglacial maximum under the second, somewhat stronger insolation peak at 409.5 kyr BP.

This proposed mechanism is supported not only by a high SST anomaly in the mid/subtropical Atlantic during early MIS 11c (M23414 and ODP Site 958; Supplementary Fig. 1) (refs. 74,75), but also by the coeval presence of cold, low-salinity sea-surface water in the Nordic Seas (MD99-2277 and M23063; Supplementary Fig. 1). Although not absolutely dated, these proxy records indicate temporal coincidence between continuous northward transport of warm water from the mid-Atlantic towards high latitudes and sustained melting in the high latitudes during early MIS 11c (refs. 75,76). Our results support model simulations[16], and isotopic data[73] that suggest the importance of prolonged, albeit moderate summer warming for achieving the full interglacial culmination of MIS 11c. This process operated in concert with the summer energy increase[77], which—integrated over the long timescale of the transition into MIS 11c (from a summer insolation energy minimum to a maximum, 436.6–414.4 kyr BP)—contributed a ~40% larger forcing than that leading into the Holocene (26.1–10.3 kyr BP). This was conducive to greater ice-mass loss and, thus, sea-level rise to (eventually) 6–13 m above present[12].

The glacial-interglacial transition from MIS 12 through T-V to MIS 11c comprised a distinct sequence of events, in which early establishment of interglacial conditions in the subtropical to mid-latitude North Atlantic in response to a weak insolation maximum at 425.6 kyr BP played a central role. This insolation maximum may have been sufficient to melt the southern margins of the extensive

MIS 12 ice sheets early in MIS 11c (Fig. 4c). Over the subsequent 15 kyr, the early interglacial conditions established in the subtropical to mid-latitude North Atlantic may have driven a strong poleward heat flux via enhanced Atlantic ocean circulation (associated with strong AMOC), which preconditioned the climate system for full deglaciation at the next insolation maximum. This then culminated in warm full-interglacial conditions at a global scale, with sea level rising to 6–13 m above present, after 410 kyr BP (refs. 11,12). Note that our analysis neither precludes the impacts of other feedbacks (e.g., vegetation-albedo feedback[78,79]) nor concludes that early North Atlantic warming at lower latitudes was unique to MIS 11c. It might be a ubiquitous feature of all terminations that simply is (virtually) indistinguishable in more rapid terminations. The long duration of T-V facilitates distinction of successive stages of the sequence, and thus provides a key case study for understanding the processes involved in glacial terminations.

## Methods

### Speleothem sampling and stable isotope measurements

Flowstone core BA7-1 was obtained from a chamber located 400 meters from the entrance of Bàsura cave in January 2014. This drilling site is 150 meters away from where cave air temperature and humidity (BA19-2) were measured[38] (Supplementary Fig. 1b). The core was halved and polished prior to sampling and analyses This study targets the 317 to 1,120 mm interval in BA7-1. A total of 1,139 powdered subsamples, each of 10–50 μg, were micro-milled at 0.05–0.5 mm intervals along the growth axis of BA7-1 (Supplementary Data 1) for stable oxygen ($\delta^{18}O$) and carbon ($\delta^{13}C$) isotope analysis. Hendy tests[40] were performed using 4 to 7 subsamples from 14 of these layers (Supplementary Fig. 3). $\delta^{18}O$ and $\delta^{13}C$ analyses were performed on a Thermo-Finnigan MAT 253 at Fujian Normal University and a Micromass IsoPrime mass spectrometers at National Taiwan Normal University. At Fujian Normal University, three standards were used, including NBS-19 ($\delta^{13}C = 1.95$ ‰, $\delta^{18}O = -2.20$ ‰), IAEA-603 ($\delta^{13}C = 2.46$ ‰, $\delta^{18}O = -2.37$ ‰), and CAI-13 (in-house standard; $\delta^{13}C = -10.73$ ‰; $\delta^{18}O = -9.46$ ‰). At National Taiwan Normal University, three standards were used, including NBS-19 ($\delta^{13}C = 1.95$ ‰, $\delta^{18}O = -2.20$ ‰), IAEA-CO1 ($\delta^{13}C = 2.49$ ‰, $\delta^{18}O = -2.4$ ‰), and MAB (in-house standard, marble from Taiwan; $\delta^{13}C = 3.4$ ‰, $\delta^{18}O = -6.88$ ‰). These standards are routinely measured for monitoring the precision and accuracy of the isotope analyses. All values are reported in per mil (‰), relative to the Vienna PeeDee Belemnite (VPDB). The reproducibility of $\delta^{18}O$ and $\delta^{13}C$ measurements was better than ± 0.12 ‰ (1σ) and ± 0.06 ‰ (1σ), respectively.

### Sr/Ca analyses in speleothem calcite

Sr/Ca compositions were measured on an inductively coupled plasma-sector field mass spectrometer Finnigan Element II using 558 powdered subsamples of 10-50 μg, which were micro-milled at 0.1–0.5 mm intervals along the BA7-1 growth axis. The 2σ reproducibility is ± 0.5% with external matrix-matched standards analysed every 4-5 samples[80]. Sr/Ca data are given in Supplementary Data 2.

### U-Th dating

U-Th chemistry[81] and dating[82] (Supplementary Data 3) were performed on a Thermo-Finnigan Neptune multi-collector inductively coupled plasma mass spectrometer using 57 subsamples of 0.4–4 g. The isotope dilution method with a triple-spike $^{229}Th$–$^{233}U$–$^{236}U$ tracer was employed to correct for instrumental fractionation and determine U and Th isotopic ratios and concentrations. For U measurement, we employed a protocol modified from Cheng et al. (2013)[82]. This protocol used a 4-step jumping mode. The $1^{st}$ step was to measure the tailing background (from $^{238}U$ and $^{235}U$) at M/Z = 233.54 on the central secondary electron multiplier (SEM). At the $2^{nd}$ step, with $^{234}U$ on the central cup, Faraday amplifiers equipped with $10^{12}$, $10^{13}$, $10^{11}$, $10^{12}$, and

$10^{10}$ Ω feedback resistors were used to simultaneously measure $^{233}U$, $^{234}U$, $^{235}U$, $^{236}U$, and $^{238}U$, respectively, with an idle time of 8 seconds due to the low response time of the $10^{13}$ Ω resistor. The 3rd step was to measure the $^{234}U$ ion beam on the SEM to earn yield correction with the one measured on the cup in the 2nd step. The 4th step was to measure the tailing background at M/Z = 235.54. The tailing background on $^{234}U$ was corrected through exponential interpolation from signals at M/Z = 233.54 (1st step) and 235.54 (4th step). While this protocol did not collect the large $^{238}U^+$ beam in Faraday cups at the 1st and 4th steps, the determined atomic $^{234}U/^{238}U$ ratios of the international standard CRM-112a and Harwell Uranite-1 (HU-1) are 5.2851 ± 0.0015 (2σ) ×$10^{-7}$ and 5.4904 ± 0.0015 (2σ) ×$10^{-7}$, consistent with reported values of 5.2852 ± 0.0015 (2σ) × $10^{-7}$ and 5.4904 ± 0.0011 (σ) × $10^{-7}$ [82], respectively. Uncertainties for the $^{230}Th$ ages, relative to 1950 C.E., are given at the two standard errors (2σ). Three layers are duplicated at depths of 849.0-, 855.0-, and 887.5-mm. A total of 54 ages were eventually used for constructing the age model using StalAge (Supplementary Fig. 4)[41], which computes the median age and the associated 95% confidence limits for each depth in the BA7-1 flowstone.

## Interpretation of Sr/Ca and δ¹³C in Bàsura cave

In-situ monitoring of cave drip water rates at BA-1901, 1902, and 1907 (Supplementary Fig. 2a) indicates that dripping points at Bàsura cave are not active year-round, exhibiting distinct seasonality with a dry-summer/wet-winter pattern (Supplementary Fig. 9). For instance, the drip rates at Site BA-1901 and 1-month-lagged rainfall amount documented in Nice meteorological station (i.e., drip water lags rainfall records by 1 month) yield a correlation coefficient of 0.65 ($n = 33$, $p < 0.1$). This correlation suggests a quick (monthly) response of drip water to climate change, which may be attributed to the thin ceiling bedrock of Bàsura cave, ranging from a few meters to approximately 50 meters. The thin bedrock and short pathways of infiltrating water make the drip water sensitive to seasonal and short-term changes outside the cave.

The cave Sr/Ca ratios measured in Bàsura cave drip water are responsive to modern rainfall changes[38], primarily reflecting the extent of prior carbonate precipitation (PCP)[44,83–85]. Dry conditions increase the residence time of the infiltrated water in the epikarst and lower the drip rates that enhance $CO_2$ degassing, thereby favouring PCP. High Sr/Ca ratios during dry periods arise from preferential removal of Ca due to PCP.

$δ^{13}C$ in Bàsura cave calcite is thought to depend on both temperature and precipitation amount[37,86]. Speleothem $δ^{13}C$ reflects the $δ^{13}C$ of the dissolved inorganic carbon of the infiltrating water, which is controlled by the $CO_2$ concentrations in the atmosphere and soil, respiration of organic matters in the soil, soil microbial activity, and PCP extent[37]. Negative carbonate $δ^{13}C$ shifts at Bàsura cave relate to increased soil microbial activity or/and vegetation density due to warmer and/or more humid climate conditions. The vegetation type in the study area changed little between glacial and interglacial periods[87], which implies that the C3 or C4 type vegetation distribution above the cave is not a key factor controlling Bàsura cave $δ^{13}C$. During glacial terminations, an ~100 ppm rise in atmospheric $CO_2$ concentrations can contribute at most a -1.3 ‰ decrease in $δ^{13}C$ (ref. 88), which can account for ~20% of the ~5 ‰ $δ^{13}C$ decrease at the end of T-V in the BA7-1. This leaves a dominant role for precipitation amount in controlling BA7-1 $δ^{13}C$. This is corroborated by the strong covariation between $δ^{13}C$ and Sr/Ca because Sr/Ca in Bàsura cave is sensitive to precipitation amount changes[38].

The Sr/Ca and $δ^{13}C$ is unlikely to be biased by seasonality changes during interglacial-glacial cycles. First, pollen assemblage data[89] indicate that the rainy seasons coincided with the winter-half-year (September-March) during both glacial and interglacial periods in the northern Mediterranean borderlands. Hence, the seasonal patterns of precipitation remained unchanged despite changing climate

boundary conditions. Second, while the reduced evaporation in glacial summers due to cooler temperatures[90] could have increased net infiltration, in our precipitation-driven Sr/Ca record, we observed a value of -0.10 in glacial MIS 12 and -0.04 in interglacial MIS 11c. This suggests that our interpretations are not biased by increases in glacial summer precipitation, because that should have led to lower Sr/Ca values in MIS 12. The strong evaporation in glacial summer could still limit the extent to which summer rainwater effectively influences the infiltration water in the Bàsura cave system. This argument is supported by pollen-based summer temperature reconstructions for Lake Ohrid, Macedonia, in which MIS 12 summer temperature is around 12-18°C, not much lower than that in modern days of ~20°C (ref. 90).

## Processes controlling speleothem δ¹⁸O in Bàsura cave and moisture sources

Speleothem calcite $δ^{18}O$ in principle reflects precipitation $δ^{18}O$, which is influenced by the $δ^{18}O$ of the oceanic moisture source, moisture trajectories, air temperature, and the amount of (winter-half-year) precipitation[91]. Today, Bàsura cave speleothem $δ^{18}O$ primarily registers recharge-weighted $δ^{18}O$ during late autumn to early spring[39]. This is because approximately 50–70% of the annual precipitation amount is lost through evapotranspiration[92], notably during hot and dry summers. In this region, the effect of air temperature on rainfall $δ^{18}O$ (+0.2 ‰/°C, refs. 93,94) commonly counterbalances the temperature-controlled $δ^{18}O$ fractionation during calcite precipitation (−0.2 ‰/°C, ref. 95). Hence, amount and source effects are left as the main controls on Bàsura cave speleothem $δ^{18}O$ through time[39].

The amount effect causes the $δ^{18}O$ of precipitation (and hence of speleothem calcite) to become more negative as the amount of precipitation increases[96]. The source water effect predominantly reflects input of low-$δ^{18}O$ meltwater into the ocean during glacial terminations, which cause a reduction in the $δ^{18}O$ of vapour derived from those surface waters, which in turn reduces both rainwater and speleothem $δ^{18}O$ at the cave location (refs. 31–33). Change in the location of the moisture source region can also influence speleothem $δ^{18}O$ values, whereby Atlantic-sourced moisture (−8.5 ‰, ref. 97) contributes more negative $δ^{18}O$ to precipitation (and to speleothem calcite) at Bàsura cave than moisture sourced from the Mediterranean Sea (−4.6 ‰, ref. 97). This balance may have changed through time, especially if/when different meltwater additions affected evaporating water in either basins.

Moisture tracking analysis[98] show that during the rainy seasons of autumn to spring (Supplementary Fig. 6b and c), over 50% and 25–30% of the moisture in the western Mediterranean is sourced from the Atlantic Ocean and the Mediterranean (including its borderlands), respectively. Among these sources, Atlantic-sourced moisture dominates during winter to early spring due to the meridional temperature contrast over the Atlantic Ocean, which strengthens westerly winds and the associated advection of moisture-laden air masses[99]. During autumn, the influence of Mediterranean-sourced moisture and local recycling processes on precipitation amount in the western Mediterranean increases, as the large thermal gradient between warm Mediterranean SST and cold air enhances evaporation and baroclinic instability[93]. Occasional summer thunderstorms or cyclogenesis originating in the Atlantic or the Gulf of Genoa may potentially result in extremely $^{18}O$-depleted rainfall due to intense Rayleigh fractionation. However, the impact of these events on long-term timescales cannot (yet) be distinguished unequivocally[93,100].

The interpretation of BA7-1 $δ^{18}O$ across the MIS 12 to MIS 11c interval is not straightforward. Seasonal rainfall patterns, location of the moisture source(s), and meltwater pulses from the Northern Hemisphere ice sheets and Alpine glaciers could have weighted differently on local precipitation $δ^{18}O$ under different orbital configurations (cf. ref. 56). Over (multi-)millennial scales, the BA7-1 $δ^{18}O$

record (Supplementary Fig. 10a) shares similarities with planktic foraminiferal $\delta^{18}$O records ($\delta^{18}O_{pf}$) from Integrated Ocean Drilling Program (IODP) Site U1385 in the North Atlantic ($r = 0.29$, $p < 0.05$; 1-kyr-resampled; Supplementary Fig. 10a)[101], from core PRGL1-4 in the northernmost western Mediterranean Sea (Gulf of Lion) ($r = 0.66$, $p < 0.001$; Supplementary Fig. 10b and c)[102], and from Ocean Drilling Program (ODP) 967 and core KC01B in the eastern Mediterranean Sea ($r = 0.72$, $p < 0.001$; Supplementary Fig. 10b and d)[103]. This supports the concept that speleothem $\delta^{18}$O from the Mediterranean borderlands (at least to a substantial extent) reflects $\delta^{18}$O changes in the moisture source areas.

## Probabilistic analysis

We used a Monte Carlo approach in MATLAB[32,104] to probabilistically evaluate the stable isotope data from Bàsura cave, and to evaluate and stack previously published SST reconstructions[21,57] on their new, radiometrically constrained chronologies as presented in this study. Input data for the Monte Carlo routine (5000 simulations) were sample ages and proxy data with their 1σ uncertainties. The chronological uncertainties were evaluated using a random walk Monte Carlo routine that employs a Metropolis–Hastings approach to reject steps in the random walk that result in age reversals[105]; i.e., it imposes a monotonic age increase with depth (cf. ref. [106]) because the data were measured in a stratigraphically coherent manner along an individual flowstone or sediment core. All realizations were then linearly interpolated on their respective radiometric time scales to produce ensembles of 5000 time series for each of the analysed records from Bàsura cave, IODP Site U1313, and core MD03-2699. Probability distributions were assessed at each time step, marking the 68% (16th–84th percentile) and 95% (2.5th–97.5th percentile) confidence intervals as well as the median (50th percentile). Ensembles of SST time series generated with the Monte Carlo approach for IODP Site U1313 and core MD03-2699 were stacked and probabilistically evaluated to generate the North Atlantic SST stack with its median value and the 68% and 95% confidence levels.

## Connection between North Atlantic sea-surface temperature and Bàsura precipitation amount

Mid-latitude Atlantic (Site U1313 and MD03-2699) SST variability correlates well with Greenland temperature changes and Atlantic ocean circulations during recent glacial-interglacial cycles[66,107,108]. Mid-latitude ocean surface cooling and warming coincide with southward migrations of cold subpolar waters and northward expansion of subtropical waters, respectively. The movement of these water masses, reflected in the position of the polar front (the boundary of warm and cold water masses), is controlled by the strength of the Atlantic surface circulation, primarily the North Atlantic Current (NAC) (cf. refs. [109,110]). Specifically, a stronger NAC and intensified sub-tropical gyre in the central-eastern Atlantic drives more salty and warmer waters into the Nordic Seas, causing a northward retreat of the polar front and high-latitude sea-ice reduction. Site U1313 (at 41°N) and MD03-2699 (at 39°N) are suitable for tracking the position of the polar front[21,57] and hence the variability of the NAC because the southernmost position of the polar front under full glacial conditions was at ~40°N (refs. [107,110]).

North Atlantic warming and a strong NAC could potentially be accompanied by increased North Atlantic storminess[111]. While the extent to which North Atlantic warming impacts the position of the mid-latitude wind belts remains a subject of debate[112], on millennial to orbital scales, both proxy records[52–54] and model simulations[55,56] generally suggest a synchronous variation in North Atlantic SST and precipitation amount patterns in the western Mediterranean. Proposed mechanisms[47] indicate that warm North Atlantic and Mediterranean seawater tends to create a significant land-sea temperature contrast, favouring the formation of local cyclones within the western Mediterranean basin. For example, increasing land-sea temperature contrasts lead to intensified baroclinic instability that favours moisture convergence and that may facilitate the formation of cyclones in the Gulf of Genoa[113,114]. These influences are particularly strong in autumn and winter[113,114], when continental temperatures drop while sea-surface temperatures remain high due to the large ocean inertia. Aligning with this, current models[115] also show that a warmer North Atlantic Ocean – and thus steeper meridional temperature gradients – could potentially strengthen and extend the winter North Atlantic jet stream further into Europe. High SST with high heat fluxes from the ocean to the atmosphere could also amplify the efficiency of moisture advection from the North Atlantic to the western Mediterranean[52,53,107]. Accordingly, high North Atlantic SSTs could favour enhanced precipitation amount in the Bàsura region.

## Transferring the Bàsura cave chronology to other palaeoclimate archives

To facilitate a quantitative analysis of the onset and climate evolution of MIS 11c, the Bàsura cave chronology based on U-Th dating and StalAge modelling was transferred to two marine sediment archives from the mid-latitude North Atlantic Ocean[21,57] at IODP Site U1313 and MD03-2699. The correlation between Bàsura $\delta^{13}$C and marine SST hinges on the relationship observed in instrumental data (Supplementary Fig. 6a), palaeoclimate data[53,54], and simulation results[47] between Atlantic Ocean conditions and Bàsura region (Mediterranean) precipitation amount. Specifically, two $^{230}$Th-based tie-points of the Bàsura record were chosen from the temporal maxima of 1-kyr filtered change rates of each of the proxy time series (Supplementary Fig. 7; cf. ref. [116]). The two control points (rate maxima) at ~425 and ~410 kyr BP could represent the climatic response to Termination V and the insolation rise during MIS 11c, respectively. The chronology of the sections beyond the two tie-points follows their original age models. Propagation of the age and sample spacing uncertainties of each tuned record follows Marino et al. (ref. [32]). Because the resolution of the marine records is around 1.5 kyr/datapoint, an additional, generous uncertainty of ± 3 kyr (2σ) was taken into account in the final results in order to avoid underestimating the uncertainties involved tuning processes. The tuned age models for Site U1313 and MD03-2699 are reported in Supplementary Data 4.

## Change-point analysis and sensitivity tests

The change points in the series are identified using the RAMPFIT and BREAKFIT algorithms[59,60]. RAMPFIT identifies two change points and the constant pretransition and post-transition levels with a ramp fitted between them. This feature allows the end point of T-V in the Atlantic SST to be determined. In the case of Bàsura $\delta^{13}$C, the BREAKFIT algorithm was employed, which identifies one change point and fits a linear slope on each side.

To assess the robustness of the identified change points, sensitivity tests were conducted by duplicating RAMPFIT and BREAKFIT tests with randomly chosen search time windows. The results are considered reliable when consistent solutions are obtained, with the timing of the identified change points varying by no more than 1 kyr when the search time window changes. Sensitivity tests indicate that our estimates of change points in Bàsura $\delta^{13}$C and Atlantic SSTs are all reliable (Supplementary Table 1). The main text refers to the RAMPFIT and BREAKFIT results obtained using a search time window of 415–435 kyr BP and 415–429 kyr BP, respectively. The reported errors were calculated using root-mean-square errors, which propagate all chronological uncertainties from RAMPFIT (or BREAKFIT) results and tuning model ages (Supplementary Table 1).

## Data availability

Supplementary Data 1 to 4 are provided with this paper.

## Code availability

Codes will be made available by the corresponding authors upon request.

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

## Acknowledgements

H.-M.H. acknowledges help from the Japan-Taiwan Exchange Association. C.-C.S. thanks the Atmosphere and Ocean Research Institute at the University of Tokyo for the visiting professor fellowship funding. The authors thank Tanguy Racine for technical help on speleothem age models and Marie-Hélène Moncel (dir. Erc lateurope) for her supports on Taiwan-France collaborations. This work is supported by the National Science and Technology Council, Taiwan, ROC, 110-2123-M-002-009 (C.-C.S.), 111-2116-M-002-022-MY3 (C.-C.S.), 111-2926-I-002-510-G (C.-C.S.), Higher Education Sprout Project of the Ministry of Education, Taiwan, ROC, 112L901001 (C.-C.S.), National Taiwan University, Taiwan, ROC, 112L894202 (C.-C.S.), Japan Society for the Promotion of Science, Japan, JP16H02235 (A.K.), 20H00193 and 23KK0013 (Y.Y.), ERC Adv grant, LATEUROPE n° 101052653 (V.M. and P.V.), NSFC42330403 (J.Y.), Climate Initiative REKLIM grant from Helmholtz Association (P.L.), AXA Research Fund (J.G.P.), Universidade de Vigo, Spain (RRO4092017), a Beatriz Galindo Fellowship (BG20/00157), and the TRIPACC project (PID2019-109653RB-I00), funded by MICIU/AEI/ 10.13039/ 501100011033 (G.M.).

## Author contributions

Conceptualization, Data Curation, and Writing – original draft: H.-M.H. Methodology and Visualization: H.-M.H. and G.M. Software: H.-M.H. and G.M. Resources: E.S., M.Z., V.M., and P.V. Investigation: H.-M.H., G.M., A.K., X.Z., X.J., H.-S.M., W.-Y.C., H.-C.T., W.-H.S., C.-H.H., and C.-C.S. Writing – review & editing: G.M., C. P.-M., C.S., Y.Y., J.Y., E.R., P.L., J.G.P, and C.-C.S. Funding acquisition: G.M., Y.Y., A.K., P.L., J.G.P., V.M., P.V., and C.-C.S. Supervision and Project administration: H.-M.H. and C.-C.S.

## Competing interests

The authors declare no competing interests.
