## [Peer Review File · Nature Communications]

Sustained North Atlantic warming drove anomalously intense MIS 11c interglacialREVIEWER COMMENTS

Reviewer #1 (Remarks to the Author):

Sustained North Atlantic warming drives anomalously intense MIS 11c interglacial by Hu et al aims to address the outstanding question that insolation forcing during MIS 11 was weak compared to other Pleistocene interglacial periods, and yet climate was warm and ice sheets were small (sea level was high) compared to those same interglacial periods. The authors address this question by generating a radiometrically-dated record of carbon and oxygen stable isotope values and Sr/Ca ratios from a cave in the northern Mediterranean region. The conclusion of this study is that warming began during a northern hemisphere summer insolation peak ~425 ka, well before to a larger northern hemisphere summer insolation peak and to peak warmth and sea level at ~410 ka (MIS 11c). The study concludes that sustained northward heat flux from the subtropics would have primed climate and northern hemisphere ice sheets to respond sensitively to the minimal insolation forcing, resulting in anomalous warmth and ice sheet retreat during MIS 11c.

The central question that this manuscript aims to address is both interesting and timely, a focus of ongoing research and discussions. By providing a radiometrically dated record from Basura cave in the North Atlantic region, this manuscript provides a strong contribution to this question, as radiometrically dated records are difficult to generate and therefore rare for this time period. The methods for producing this record seem appropriate and well executed. The manuscript also uses correlation between records to transfer the radiometric-based age model from Basura cave to marine and lacustrine records in this region. This approach seems less well executed, and has large uncertainties associated with decisions on the location of tie points (different researchers approach this differently), uncertainties that are not represented in the manuscript, and that could change the conclusions regarding the timing of key events in the records by thousands of years or more. I would therefore suggest that the portion of the manuscript leaning on the transfer of these ages be removed, or heavily modified to incorporate either the original age-depth models or the very large uncertainties associated with the tie-point approach. This study concludes with an interesting discussion of the mechanisms that would have operated in the climate system to cause the sequence of events reconstructed by this new radiometric record and by the Red Sea sea-level record. This discussion requires some more detailed explanation and support, described below. This manuscript provides an important new record to address fundamental questions about the interplay of glaciers and climate during periods of warming and prolonged warmth. This manuscript will be strengthened to publishable quality with 1. some modifications relating to transferring age models via tie points and the discussion dependent on that approach, 2. with the addition of details about the climate mechanisms operating during this time, and 3. With minor modifications described below.

Below I provide more detailed explanations of the suggested modifications, in addition to some minor suggestions:

Parenthetical statements to provide information about opposites are incredibly confusing to understand and distracting to read (see this Eos article: Parentheses Are (Are Not) for References and Clarification (Saving Space)

<https://agupubs.onlinelibrary.wiley.com/doi/abs/10.1029/2010EO450004>).

Please remove these parenthetical statements from throughout the manuscript and instead use phrasing such as:

Original: Heavy (light) $\delta^{13}\text{C}$ indicates cold and/or dry (warm and/or humid) climatic

conditions with low (high) soil bioactivity or enhanced (limited) PCP.

Revised: Heavy $\delta^{13}\text{C}$ indicates cold and/or dry climatic conditions with low soil bioactivity or enhanced PCP, with opposite conditions indicated by light $\delta^{13}\text{C}$.

Second revised option: Heavy $\delta^{13}\text{C}$ indicates cold and/or dry climatic conditions with low soil bioactivity or enhanced PCP, and vice versa for light $\delta^{13}\text{C}$.

There is a terrestrial temperature record for Greenland during Pleistocene interglacials that would be useful to cite in addition to the deVernal pollen record, the marine records, and the glacial geologic reconstructions when discussing Greenland Ice Sheet response to this and other interglacials: Cluett & Thomas 2021 (doi.org/10.1073/pnas.2022916118). This record also addresses questions regarding the mechanisms that could have caused Greenland Ice Sheet retreat during MIS11.

Line 114: 'two culmination of interglacial conditions' this implies an ending to me. Do the authors instead perhaps intend to say 'two periods of peak interglacial conditions' or something like that?

Line 120: "The Basura chronology is, thus, transferred to North Atlantic temperature records to quantitatively examine the timing and magnitude of MIS 11c warming in this critical region for glacial-interglacial climate change"-how did the authors do this? Supp text? Methods? (please add ref to where this info is)

Line 751: red and blue shading (not contours)

Supp Fig 6a, where are the arrows derived from? Cite this. Are the arrows representative of modern reanalysis data or climate model data of an older interglacial or something else?

Supp Fig 6b: would be easier to view range of values through this time period if displayed as box plots, showing interquartile range & whiskers about median values for each month, rather than bars

Line 123 & throughout: since both precipitation $\delta^{18}\text{O}$ and precipitation amount are referred to throughout this paper, it'd be most clear to state "precipitation amount" when the authors mean that variable, to distinguish from precipitation $\delta^{18}\text{O}$. Easy to add!

Line 125: Precipitation amount and seasonality at Basura cave?

Paragraph beginning at Line 140: this is confusing to follow. Can the sentences that describe this approach be shortened to include fewer lists, perhaps with the steps being laid out first, then discussed in terms of the different records. I'm confused about the entire paragraph, but especially about the part beginning "we consider that MIS 11c". Some additional aspects that are unclear: are these ages based on the radiometric chronologies provided in this study? Why is it important to use these records to identify the magnitude and duration of the interglacial when the authors have a radiometrically dated speleothem? Why does the Holocene have to come into play-can't terminations be different durations and start/end at different temperatures, and can't the magnitude and duration of the termination be determined from the records themselves, rather than via a comparison with another termination?

The discussion of the timing of onset of the termination among these records is circular, when the chronologies of all of them are tied to Basura. For example, I was trained to use the middle of rapid transitions as tie points between records, rather than peak values, and taught that I should never use tie points that are at the end of a record or in a gap of a record

(eg tie point at ~455 ka in Fig S8). So, if I were to do this analysis, I would get different correlation-based age models for the marine and lake records than the authors did. And I would have different conclusions about the timing of onset of the interglacial period.

Is there a more objective method to develop independent age models for these records? Can the authors instead use the age-depth models from the original publications? If not, due to the circular nature of this aspect of the study, I suggest that the authors leave the timing of interglacial onset based on the marine and lake records out of the manuscript.

line 189: Is this statement based on data shown in Fig 2, from the Basura record? Add reference to the relevant figure and/or add text indicating “the basura record indicates that during MIS12...”

Line 196: By what mechanism would this warmth cause enhanced heat advection to high latitudes? Can the authors describe this in more detail?

Line 216: “We demonstrate that this insolation maximum sufficed to melt the southern margins of the extensive MIS 12 ice sheets early in MIS 11c.” This statement seems a bit strong: since I don’t see calculations that indicate this was demonstrated in the manuscript, nor was this type of study the focus of data generation/comparison in this manuscript- perhaps reword or clarify how this manuscript demonstrates that fact?

Line 218: What is a ‘heat-pump feedback’? Can the authors clarify this and provide more detail?

Reviewer #2 (Remarks to the Author):

This manuscript presents a new radiometrically-dated palaeoclimate record for Marine Isotope Stage 12, Termination V and Marine Isotope Stage 11c. The authors use a well dated flowstone from a cave in northern Italy to closely examine the links between solar insolation and global ice volume with climate changes in the Mediterranean and the broader North Atlantic region.

The study addresses a longstanding puzzle in palaeoclimate: MIS 11 was the longest and one of the warmest interglacial periods of the last 800,000 years, yet it occurred at a time when boreal summer insolation forcing was weak. Therefore, there is a clear mismatch between the forcing and the climate response to this forcing. How was it possible for a relatively small change in solar insolation to initiate the melting of the largest ice sheets of the last 800,000 years and why was the subsequent interglacial period so prolonged and unusually warm? Answering these questions is fundamental to our understanding of drivers of ice ages and global climate.

The novelty of this study is that, unlike previous reconstructions, it has the necessary dating control and proxy resolution to more precisely correlate the climate signals with the insolation and ice volume changes. The authors find that a weak insolation maximum at ~425 ka led to the onset of interglacial conditions at subtropical and mid-latitudes in the North Atlantic region and to the melting of the southern margins of the northern hemisphere ice sheets. The persistence of warm conditions over the subsequent millennia and the export of heat towards higher latitudes preconditioned the climate system to be more sensitive to

the next rise in solar insolation which led to a full deglaciation. This study is likely to be of interest for a wide range of palaeoclimate and climate scientists.

The proxy data presented and the U-Th dating is of high quality, data interpretation is robust, and conclusions are well supported by the data presented. I only have some minor suggestions for improving the manuscript, as described below:

1. Some more information about the physical setting of the flowstone in the cave would be useful. In previous publications by the main author, the cave was described as being 1 km long, with stable temperatures and high humidity. But where in the cave was the flowstone corrected relative to where the temperature and humidity measurements were made, and relative to the cave entrance?
2. While most of the precipitation is in the winter months, are the cave dripping points active through the year or only in the winter? In other words, how fast is the cave responding to outside rainfall? Also, is it likely that cave recharge will remain biased towards winter season during glacial periods as well as interglacials? Could the changes in stable isotopes and PCP be driven by shifts in the annual distribution of precipitation on glacial-interglacial timescales?
3. Given the importance of the age model for conclusions, could you demonstrate that the results are similar if you choose a different age model algorithm? A comparison between outputs from StalAge and COPRA or OxCal (or similar algorithms) would strengthen your argument.
4. The stable isotopes measurements were made on two different instruments. What measures were taken to ensure there are no systematic offsets between these instruments?
5. The manuscript doesn't meet the latest IUPAC guidelines for reporting stable isotope ratio measurements (see Coplen, T.B., 2011, Rapid Commun. Mass Spectrom., 25, 2538). Please revise the methods section to include: a) what standards were used in addition to NBS-19 and what were their isotopic values, b) based on what standard was the reproducibility of the isotopic measurements established, c) italicise delta symbol and leave space between number and permil symbol.

Kind regards,

Vasile Ersek

We appreciated the reviewers' constructive comments and suggestions to improve the quality of this research. Below, we list the review comments in blue, followed by our responses in black.

Reviewer #1

Sustained North Atlantic warming drives anomalously intense MIS 11c interglacial by Hu et al aims to address the outstanding question that insolation forcing during MIS 11 was weak compared to other Pleistocene interglacial periods, and yet climate was warm and ice sheets were small (sea level was high) compared to those same interglacial periods. The authors address this question by generating a radiometrically-dated record of carbon and oxygen stable isotope values and Sr/Ca ratios from a cave in the northern Mediterranean region. The conclusion of this study is that warming began during a northern hemisphere summer insolation peak ~425 ka, well before to a larger northern hemisphere summer insolation peak and to peak warmth and sea level at ~410 ka (MIS 11c). The study concludes that sustained northward heat flux from the subtropics would have primed climate and northern hemisphere ice sheets to respond sensitively to the minimal insolation forcing, resulting in anomalous warmth and ice sheet retreat during MIS 11c.

The central question that this manuscript aims to address is both interesting and timely, a focus of ongoing research and discussions. By providing a radiometrically dated record from Basura cave in the North Atlantic region, this manuscript provides a strong contribution to this question, as radiometrically dated records are difficult to generate and therefore rare for this time period. The methods for producing this record seem appropriate and well executed. The manuscript also uses correlation between records to transfer the radiometric-based age model from Basura cave to marine and lacustrine records in this region. This approach seems less well executed, and has large uncertainties associated with decisions on the location of tie points (different researchers approach this differently), uncertainties that are not represented in the manuscript, and that could change the conclusions regarding the timing of key events in the records by thousands of years or more. I would therefore suggest that the portion of the manuscript leaning on the transfer of these ages be removed, or heavily modified to incorporate either the original age-depth models or the very large uncertainties associated with the tie-point approach. This study concludes with an interesting discussion of the mechanisms that would have operated in the climate system to cause the sequence of events reconstructed by this new radiometric record and by the Red Sea sea-level record. This discussion requires some more detailed explanation and support, described below. This manuscript provides an important new record to address fundamental questions about the interplay of glaciers and climate during periods of warming and prolonged warmth. This manuscript will be strengthened to publishable quality with 1. some modifications relating to transferring age models via tie points and the discussion dependent on that approach, 2. with the addition of details about the climate mechanisms operating during this time, and 3. With minor modifications described below.

We thank Reviewer#1 for her/his summary in which the timeliness of our study is highlighted, notably how our radiometrically dated speleothem records contribute to address the so-called MIS 11c paradox, with

intensely developed interglacial conditions (high sea level) at a time of relatively weak insolation forcing. We appreciate the suggestions made to improve our study and the comments, which were all addressed in our revision and which helped to strengthen our analysis and conclusions.

R1.1. Below I provide more detailed explanations of the suggested modifications, in addition to some minor suggestions:

Parenthetical statements to provide information about opposites are incredibly confusing to understand and distracting to read (see this Eos article: [Parentheses Are \(Are Not\) for References and Clarification \(Saving Space\)](https://agupubs.onlinelibrary.wiley.com/doi/abs/10.1029/2010EO450004))

<https://agupubs.onlinelibrary.wiley.com/doi/abs/10.1029/2010EO450004>). Please remove these parenthetical statements from throughout the manuscript and instead use phrasing such as:

Original: Heavy (light) $\delta^{13}\text{C}$ indicates cold and/or dry (warm and/or humid) climatic conditions with low (high) soil bioactivity or enhanced (limited) PCP.

Revised: Heavy $\delta^{13}\text{C}$ indicates cold and/or dry climatic conditions with low soil bioactivity or enhanced PCP, with opposite conditions indicated by light $\delta^{13}\text{C}$.

Second revised option: Heavy $\delta^{13}\text{C}$ indicates cold and/or dry climatic conditions with low soil bioactivity or enhanced PCP, and vice versa for light $\delta^{13}\text{C}$.

We have revised those statements in which we used parenthetical statements to provide information about opposites. We now use both the options suggested by Reviewer#1 (e.g., Lines 94-98) as follows: “*High $\delta^{13}\text{C}$ indicates cold and/or dry climatic conditions with low soil bioactivity or enhanced PCP, with opposite conditions indicated by low $\delta^{13}\text{C}$. Sr/Ca is also tied to the PCP extent at Bāsura cave³⁸. High Sr/Ca reflects dry conditions with long karst water residence times and enhanced PCP, and vice versa for low Sr/Ca.*”

R1.2. There is a terrestrial temperature record for Greenland during Pleistocene interglacials that would be useful to cite in addition to the deVernal pollen record, the marine records, and the glacial geologic reconstructions when discussing Greenland Ice Sheet response to this and other interglacials: Cluett & Thomas 2021 (doi.org/10.1073/pnas.2022916118). This record also addresses questions regarding the mechanisms that could have caused Greenland Ice Sheet retreat during MIS11.

We thank Referee#1 for calling this study to our attention. We refer to it in our revised manuscript (Lines 207 and 220), where we cite North Atlantic records in which interglacial climates developed early in MIS 11c. In addition, the evidence of prolonged, moderate summer warming in southern Greenland during MIS 11c fits well our discussion “*Our results support model simulations¹⁶ and isotopic data^{Cluett & Thomas, 2021} that have suggested the importance of prolonged, albeit moderate summer warming for achieving the full interglacial culmination of MIS 11c.*” (Line 217-218). The study is also referred to earlier in the manuscript (Line 206) when mentioning the records that support and strong AMOC-driven northward heat transport during the first half of MIS 11c.

R1.3. Line 114: ‘two culmination of interglacial conditions’ this implies an ending to me. Do the authors instead perhaps intend to say ‘two periods of peak interglacial conditions’ or something like that?

We agree and revised this statement accordingly: “*The two broad $\delta^{13}\text{C}$ and Sr/Ca minima in BA7-1 indicate two periods of peak interglacial conditions, which occurred at 424–418 kyr BP and at 412–402 kyr BP. Their onsets coincided within uncertainties (± 2 to 3 kyr, 2σ) with the MIS 11c boreal summer insolation peaks¹⁴ at 425.6 and 409.5 kyr BP (Figure 3a).*” (Lines 114-116).

R1.4. Line 120: “The Bàsura chronology is, thus, transferred to North Atlantic temperature records to quantitatively examine the timing and magnitude of MIS 11c warming in this critical region for glacial-interglacial climate change”-how did the authors do this? Supp text? Methods? (please add ref to where this info is)

Corrected. We have referred this part to Methods (Line 121).

R1.5. Line 751: red and blue shading (not contours)

Corrected.

R1.6. Supp Fig 6a, where are the arrows derived from? Cite this. Are the arrows representative of modern reanalysis data or climate model data of an older interglacial or something else?

Corrected, the relevant statement has been revised as follows: “*Locations of Bàsura cave and marine sediment archives (core MD03-2699, IODP Site U1313) are also indicated, and the arrows indicate the dominant westerly wind directions based on modern reanalysis data (NCAR/NCEP 20th century reanalysis v3) during 1950-2008 C.E. in this region³⁸.*” (Lines 805-807)

R1.7. Supp Fig 6b: would be easier to view range of values through this time period if displayed as box plots, showing interquartile range & whiskers about median values for each month, rather than bars

Corrected.

R1.8. Line 123 & throughout: since both precipitation d18O and precipitation amount are referred to throughout this paper, it'd be most clear to state “precipitation amount” when the authors mean that variable, to distinguish from precipitation d18O. Easy to add!

Corrected. We now use “precipitation amount” throughout this manuscript.

R1.9. Line 125: Precipitation amount and seasonality at Basura cave?

Corrected.

R1.10. Paragraph beginning at Line 140: this is confusing to follow. Can the sentences that describe this approach be shortened to include fewer lists, perhaps with the steps being laid out first, then

discussed in terms of the different records. I'm confused about the entire paragraph, but especially about the part beginning "we consider that MIS 11c".

The paragraph describes the approach(es) that we use to determine the onset of MIS 11c in the North Atlantic records that were targeted by our study, namely Integrated Ocean Drilling Program (IODP) Site U1313 (41°00'N, 32°57'W, 3,426 m water depth) and MD03-2699 (39°02'N, 10°40'W, 1,865 m water depth). We have revised (and partly restructured) this paragraph to improve its clarity.

Below we provide a brief description of the two approaches:

- Approach I. We applied a change-point analysis method based on the BREAKFIT algorithm (Mudelsee, 2009). This analysis is used to determine when the SST time series "breaks" at the end of the termination and takes that as the timing of the onset of interglacial conditions. Note that a similar approach (named RAMPFIT, Mudelsee, 2000) was applied to the Bāsura $\delta^{13}\text{C}$ record to statistically determine the onset of MIS 11c.
- Approach II. We used the earliest age at which SST for IODP Site U1313 and core MD03-2699 exceed – considering the associated uncertainties – the minimum Holocene SST at the same locations and took that as the onset of MIS 11c. In other words, our analysis detects when the lower (i.e., colder) 95% confidence limit in each record exceeds the minimum temperature recorded at the same location during the Holocene.

In our initial submission these approaches were presented in a reversed order, while we feel that it is clearer for the reader to present first the results of the change-point analysis, which statistically detects the onset of the interglacial conditions solely from the structure of the records, with no underlying assumptions about the relationship with the Holocene as it is the case for the threshold approach. See lines 142-161 in our revised manuscript.

R1.11. Some additional aspects that are unclear: are these ages based on the radiometric chronologies provided in this study? Why is it important to use these records to identify the magnitude and duration of the interglacial when the authors have a radiometrically dated speleothem? Why does the Holocene have to come into play-can't terminations be different durations and start/end at different temperatures, and can't the magnitude and duration of the termination be determined from the records themselves, rather than via a comparison with another termination?

We have addressed these unclear points in the revised manuscript as follows:

- (1) Are these ages based on the radiometric chronologies provided in this study? The discussed ages are based on the radiometric chronologies provided in this study, that is, the radiometric ages of Bāsura cave. We have added this information in Lines 145-146. "...Based on the radiometrically constrained age models...".
- (2) Why is it important to use these records to identify the magnitude and duration of the interglacial when the authors have a radiometrically dated speleothem? We evaluated these to show that the marine and lacustrine records largely relied on the assumed relationship between orbital forcings and proxy records (e.g., $\delta^{18}\text{O}$) with a few

tephra datings, which may give some initial sense of chronology but then prevents an independent evaluation of proxy response to orbital forcings. Instead, using the radiometrically dated Bāsura chronology can avoid this circularity. We have expressed this perspective in Lines 122-124: “*This portrays the nature of the climate responses to the relatively weak insolation changes during MIS 11c^{14,15} in this critical region for glacial-interglacial climate change²⁶, on an absolutely constrained, radiometric timescale*”.

(3) Why does the Holocene have to come into play—can't terminations be different durations and start/end at different temperatures, and can't the magnitude and duration of the termination be determined from the records themselves, rather than via a comparison with another termination?

The Past Interglacial Working Group of PAGES (PAGES, 2016) evaluates various criteria to determine the onset and duration of an interglacial, eventually using sea level as the preferred metric. A more recent study (Köhler & van de Wal, 2020) defines interglacials on the basis of the “...lack of substantial northern hemispheric land ice outside of Greenland...”. This is also a threshold approach, because it used an ice-sheet configuration similar to the present to define an interglacial climate. However, such a definition is not suitable for MIS 11c, given that a considerable volume of land ice (sea-level equivalent of –40 m) persisted on the Northern Hemisphere during the first half of MIS 11c, when subtropical and higher North Atlantic latitudes had already warmed to interglacial climate conditions (as concluded by our study).

We therefore defined the onset and duration of MIS 11c based on two independent criteria, namely a change-point analysis and a threshold analysis of the targeted time series (see response to point **R1.10** above). For the latter, we chose the Holocene threshold because both MIS 11c and the Holocene occurred under similar (low eccentricity) orbital configurations (Laskar et al., 2011; Yin et al., 2015). In addition, Holocene spatial patterns (Cartapanis et al., 2022) are documented more precisely than for any other interglacial period because chronologies can be firmly constrained (by directly dating palaeoclimate archives, e.g., with radiocarbon) and because a plethora of records is available (Kaufman et al., 2020).

R1.12. The discussion of the timing of onset of the termination among these records is circular, when the chronologies of all of them are tied to Basura. For example, I was trained to use the middle of rapid transitions as tie points between records, rather than peak values, and taught that I should never use tie points that are at the end of a record or in a gap of a record (eg tie point at ~455 ka in Fig S8). So, if I were to do this analysis, I would get different correlation-based age models for the marine and lake records than the authors did. And I would have different conclusions about the timing of onset of the interglacial period. Is there a more objective method to develop independent age models for these records? Can the authors instead use the age-depth models from the original publications? If not, due to the circular nature of this aspect of the study, I suggest that the authors leave the timing of interglacial onset based on the marine and lake records out of the manuscript.

The reviewer's comments made us re-think the approach, and we have modified it as follows:

1. We have now sought a more objective method in transferring Bāsura age to marine archives. First, we applied a 1-kyr Gaussian filter to each proxy record and computed their change rates

using their original chronologies based on 5000 times Monte-Carlo simulated results. The tie-points were selected at the temporal change rate maxima in both Bàsura $\delta^{13}\text{C}$ and the marine records. In the revised manuscript, we thus selected two tie points at ~426 and ~411 ka. At around these timings, the SST change rates in U1313 and MD03-2699 and the change rate in Bàsura $\delta^{13}\text{C}$ all display clear temporal maxima, which likely reflect climatic responses to Termination V and the second (stronger) insolation peak in MIS 11c, respectively. For sections beyond the two tie-points, we applied the original (published) age models of all records. See Lines 402-410 and new Supplementary Figure 7.

2. Next, we added additional ± 3 kyr errors (2-sigma) in the final tuning error propagation, as a cautious approach to avoid underestimating tuning uncertainties (Lines 408-410). We chose this number because the resolution of the marine records is around 1.5 kyr/datapoint or higher. To applied an additional ± 3 kyr (2-sigma) errors could represent the worst case.
3. Beyond the tuning, we also conducted the change-point and Holocene threshold analyses on the marine records using their original age model to detect the onset of MIS 11c. These results agree with those from the records on our new Bàsura $\delta^{13}\text{C}$ -based chronologies (new Table 1). See Lines 162-171.
4. Finally, we followed this reviewer's suggestion and only used the breaking age from Bàsura $\delta^{13}\text{C}$ (423.1 ± 1.3 kyr BP) in further discussion on climate mechanisms (e.g., Lines 169-171; 177-178). Doing so, we avoid any circularity that may have been perceived in the original manuscript.

In this revision, please note that we have excluded the analysis of Lake Ohrid, given that applying RAMPFIT to data with asymmetrical temperature uncertainties is inappropriate. This decision did not alter any of our conclusions.

R1.13. line 189: Is this statement based on data shown in Fig 2, from the Basura record? Add reference to the relevant figure and/or add text indicating “the basura record indicates that during MIS12...”

We thank Reviewer#1 for calling our attention to this statement, which summarises the prominently cold and dry climate conditions in the Mediterranean region during glacial MIS 12 based on previous work with pollen data from the wider Mediterranean and Iberian Margin region. Notably, these existing lines of evidence agree with the presence of a hiatus in Bàsura speleothems. We have now added the relevant references 25, 44, and 45 (listed below) and clarified this statement (Lines 192-194 in our revised manuscript) to: “*During late glacial MIS 12, peak cold and dry conditions developed in the Mediterranean region^{25, 44, 45}, which is consistent with the interruption of speleothem growth (hiatus) of BA7-1 (Figure 2c)*”.

25. Sassoon, D., Lebreton, V., Combourieu-Nebout, N., Peyron, O., & Moncel, M. H. (2023). Palaeoenvironmental changes in the southwestern Mediterranean (ODP site 976, Alboran sea) during the MIS 12/11 transition and the MIS 441

11 interglacial and implications for hominin populations. *Quaternary Science Reviews*, 304, 108010. 442

<https://doi.org/10.1016/j.quascirev.2023.108010>

44. Koutsodendris, A. et al. (2023). Atmospheric CO₂ forcing on Mediterranean biomes during the past 500 kyrs. *Nature Communications*, 14(1), 1664. <https://doi.org/10.1038/s41467-023-37388-x>

45. Sánchez Goñi, M. F. et al. (2016). Climate changes in south western Iberia and Mediterranean Outflow variations during two contrasting cycles of the last 1 Myrs: MIS 31-MIS 30 and MIS 12-MIS 11. *Global and Planetary Change*, 136, 18–29. <https://doi.org/10.1016/j.gloplacha.2015.11.006>.

R1.14. Line 196: By what mechanism would this warmth cause enhanced heat advection to high latitudes? Can the authors describe this in more detail?

We revised this part of the discussion, by detailing the mechanism that may explain how early-MIS 11c development of full interglacial conditions in the low-to-middle latitude North Atlantic played a key role in the subsequent transition to “global” interglacial conditions. This focuses on strong northward heat transport in the North Atlantic associated with a vigorous Atlantic Meridional Ocean Circulation (AMOC) during MIS 11c, which has been proposed in the literature based on palaeoceanographic time series as well as simulations in a coupled general circulation model (e.g., Berger & Wefer, 2003; Dickson et al., 2009; Voelker et al., 2010; Kandiano et al., 2012; Rachmayani et al., 2017; Robinson et al., 2017; Galaasen et al., 2020; Cluett and Thomas, 2020):

The relevant text has been revised as follows: “*The emergence of warm and humid conditions in the subtropical and mid-latitude North Atlantic from 423.1 ± 1.3 kyr BP coincided with the onset of an interglacial mode of the Atlantic Meridional Overturning Circulation (AMOC), whereby North Atlantic Deep Water ventilated the deep Atlantic Ocean*⁷². A coupled general circulation model⁷³ simulates a vigorous AMOC during MIS 11c, leading to anomalously strong northward heat transport from the subtropical latitudes. This picture agrees with warming documented early in MIS 11c in the eastern North Atlantic⁶⁷, further north at Eirik Drift¹⁷, and in southern Greenland⁷⁴. A strong AMOC⁷³ since early MIS 11c sustained protracted (~15 kyr) warming in northern high latitudes, which has been proposed to be key to the extensive Greenland ice sheet reduction that sets MIS 11c apart from other middle to late Pleistocene interglacial periods¹⁶. These climate developments preconditioned the Earth system for reaching the intense MIS 11c interglacial maximum under the second, somewhat stronger insolation peak at 409.5 kyr BP”, see Line 200-210 of our revised manuscript.

R1.15. Line 216: “We demonstrate that this insolation maximum sufficed to melt the southern margins of the extensive MIS 12 ice sheets early in MIS 11c.” This statement seems a bit strong: since I don’t see calculations that indicate this was demonstrated in the manuscript, nor was this type of study the focus of data generation/comparison in this manuscript-perhaps reword or clarify how this manuscript demonstrates that fact?

Agreed – we have revised this statement accordingly: “This insolation maximum may have been sufficient to melt the southern margins of the extensive MIS 12 ice sheets early in MIS 11c”, see Lines 226-227 in our revised manuscript.

R1.16. Line 218: What is a ‘heat-pump feedback’? Can the authors clarify this and provide more detail?

We revised this statement in the interest of clarity (cf. our response above [R1.14]) *“Over the subsequent 15 kyr, the early interglacial conditions established in the subtropical to mid-latitude North Atlantic may have driven a strong poleward heat flux via enhanced Atlantic ocean circulation (associated with strong AMOC), which preconditioned the climate system for full deglaciation at the next insolation maximum.”* (Lines 227-230).

References:

- Berger, W. H., & Wefer, G. (2003). On the dynamics of the ice ages: Stage-11 paradox, mid-brunhes climate shift, and 100-ky cycle. In *Geophysical Monograph Series* (Vol. 137, Issue March, pp. 41–59).
- Cartapanis, O., Jonkers, L., Moffa-Sanchez, P., Jaccard, S. L., & de Vernal, A. (2022). Complex spatio-temporal structure of the Holocene Thermal Maximum. *Nature Communications*, 13(1).
- Cluett, A. A., & Thomas, E. K. (2021). Summer warmth of the past six interglacials on Greenland. *Proceedings of the National Academy of Sciences*, 118(20), 1–6.
- Dickson, A. J. et al. (2009). Oceanic forcing of the Marine Isotope Stage 11 interglacial. *Nature Geoscience*, 2(6), 428–433.
- Galaasen, E. V. et al. (2020). Interglacial instability of North Atlantic Deep Water ventilation. *Science*, 367(6485), 1485–1489.
- Kandiano, E. S. et al. (2012). The meridional temperature gradient in the eastern North Atlantic during MIS 11 and its link to the ocean-atmosphere system. *Palaeogeography, Palaeoclimatology, Palaeoecology*, 333–334, 24–39.
- Kaufman, D. S., & Broadman, E. (2023). Revisiting the Holocene global temperature conundrum. *Nature*, 614(7948), 425–435.
- Köhler, P., & van de Wal, R. S. W. (2020). Interglacials of the Quaternary defined by northern hemispheric land ice distribution outside of Greenland. *Nature Communications*, 11(1), 1–10.
- Laskar, J., Fienga, A., Gastineau, M., & Manche, H. (2011). La2010: a new orbital solution for the long-term motion of the Earth. *Astronomy & Astrophysics*, 532, A89.
- Mudelsee, M. (2009). Break function regression. *The European Physical Journal Special Topics*, 174(1), 49–63.
- Mudelsee, M. (2009). Break function regression. *The European Physical Journal Special Topics*, 174(1), 49–63.
- Past Interglacials Working Group of PAGES. (2016). Interglacials of the last 800,000 years. *Reviews of Geophysics*, 54(1), 162–219.
- Rachmayani, R., Prange, M., Lunt, D. J., Stone, E. J., & Schulz, M. (2017). Sensitivity of the Greenland Ice Sheet to Interglacial Climate Forcing: MIS 5e Versus MIS 11. *Paleoceanography*, 32(11), 1089–1101.
- Robinson, A., Alvarez-Solas, J., Calov, R., Ganopolski, A., & Montoya, M. (2017). MIS-11 duration key to disappearance of the Greenland ice sheet. *Nature Communications*, 8, 1–7.
- Voelker, A. H. L. et al. (2010). Variations in mid-latitude North Atlantic surface water properties during the mid-Brunhes (MIS 9-14) and their implications for the thermohaline circulation. *Climate of the Past*, 6(4), 531–552.
- Yin, Q., & Berger, A. (2015). Interglacial analogues of the Holocene and its natural near future. *Quaternary Science Reviews*, 120, 28–46.

Reviewer #2:

This manuscript presents a new radiometrically-dated palaeoclimate record for Marine Isotope Stage 12, Termination V and Marine Isotope Stage 11c. The authors use a well dated flowstone from a cave in northern Italy to closely examine the links between solar insolation and global ice volume with climate changes in the Mediterranean and the broader North Atlantic region.

The study addresses a longstanding puzzle in palaeoclimate: MIS 11 was the longest and one of the warmest interglacial periods of the last 800,000 years, yet it occurred at a time when boreal summer insolation forcing was weak. Therefore, there is a clear mismatch between the forcing and the climate response to this forcing. How was it possible for a relatively small change in solar insolation to initiate the melting of the largest ice sheets of the last 800,000 years and why was the subsequent interglacial period so prolonged and unusually warm? Answering these question is fundamental to our understanding of drivers of ice ages and global climate.

The novelty of this study is that, unlike previous reconstructions, it has the necessary dating control and proxy resolution to more precisely correlate the climate signals with the insolation and ice volume changes. The authors find that a weak insolation maximum at ~425 ka led to the onset of interglacial conditions at subtropical and mid-latitudes in the North Atlantic region and to the melting of the southern margins of the northern hemisphere ice sheets. The persistence of warm conditions over the subsequent millennia and the export of heat towards higher latitudes preconditioned the climate system to be more sensitive to the next rise in solar insolation which led to a full deglaciation. This study is likely to be of interest for a wide range of palaeoclimate and climate scientists.

The proxy data presented and the U-Th dating is of high quality, data interpretation is robust, and conclusions are well supported by the data presented. I only have some minor suggestions for improving the manuscript, as described below:

We thank Dr. Vasile Ersek for his constructive comments and for emphasizing the novelty of our study and the high quality (and added value) of the radiometrically dated speleothem proxy data that we present.

R2.1. Some more information about the physical setting of the flowstone in the cave would be useful. In previous publications by the main author, the cave was described as being 1 km long, with stable temperatures and high humidity. But where in the cave was the flowstone corrected relative to where the temperature and humidity measurements were made, and relative to the cave entrance?

In Figure **R2.1**, we present a map of the interior of Bàsura cave with the location of flowstone core BA7-1 used in this study. The figure also shows the locations of drip water monitoring sites BA-1901, BA-1902, and BA-1907. Temperature and humidity data measured close to BA-1902 were reported in Hu et al. (2022), while they are not available for BA-1901. BA7-1 and BA-1902 are located 100 m away from the cave entrance. Note that the current cave entrance of Bàsura cave is artificial, and was excavated in ~1950 C.E, while the location of the natural, ancient entrance is unknown. We have added the information in the Lines 240-241, with an update of Supplementary Figure 1.

Figure R2.1. Locations of flowstone BA7-1, and three drip water monitoring sites BA-1901, BA -1902, and BA -1907.

R2.2. While most of the precipitation is in the winter months, are the cave dripping points active through the year or only in the winter? In other words, how fast is the cave responding to outside rainfall? Also, is it likely that cave recharge will remain biased towards winter season during glacial periods as well as interglacials? Could the changes in stable isotopes and PCP be driven by shifts in the annual distribution of precipitation on glacial-interglacial timescales?

Monitoring shows that dripping points in Bâsura cave are not active year-round. Figure **R2.2** illustrates the drip rate at the three monitoring sites (cf. Figure **R2.1** for their location) in Bâsura cave from 2019 to 2021 C.E., with Site BA-1901 situated in the chamber where flowstone core BA7-1 was collected. The drip rates at all sites exhibit distinct seasonality, generally following a dry-summer/wet-winter pattern. Upon comparison of the drip rates with modern rainfall time series, we found a high degree of correlation, notably for a 1-to-2-month lag (Table **R2.1**). For instance, at Site BA-1901, a correlation coefficient of 0.65 between drip rates and 1-month lagged rainfall records at the Nice meteorological station, i.e., drip rate lags behind rainfall records by 1 month. This correlation suggests a fairly fast response of drip water to climate change, which may be attributed to the thin bedrock overburden of Bâsura cave, ranging from a few meters to approximately 50 meters. The thin bedrock and short pathways of infiltrating water make the drip water sensitive to seasonal and short-term changes outside the cave. We have added the related statement for clarity in Lines 271-279.

Figure R2.2. Drip rates measured at Site BA-1901, BA-1902, and BA-1907. The monitoring period was from April, 2019 to March, 2021, with a 2-months (February and March, 2020) interruption due to the COVID-19 pandemic.

Table R2.1. Lagging correlation coefficients between measured drip rates and precipitation amount.

	Site-1901	Site-1902	Site-1907
No lags	0.48	0.02	-0.06
dripwater lagging 1 month	0.65	0.61	0.58
dripwater lagging 2 month	0.17	0.51	0.06
dripwater lagging 3 month	0.02	0.35	0.02

Boldfaced numbers indicate that the statistical significance (*p*-value) of the R value is < 0.1.

Water infiltrates into Bàsura cave primarily during the winter half-year, while high evaporation results in no net infiltration during summer. The Sr/Ca and $\delta^{13}\text{C}$ data primarily reflect rainfall changes during the winter-half-year (Hu et al., 2022). Pollen assemblage data (e.g., Camuera et al., 2022) indicate that the rainy season (like today) coincided with the winter-half-year (September-March) during both glacial and interglacial periods in the northern Mediterranean region. That is, the seasonal patterns of precipitation remained unchanged despite changing climate boundary conditions. Therefore, the Sr/Ca and $\delta^{13}\text{C}$ is not likely to be biased by seasonality changes during glacial-interglacial cycles.

We also consider that reduced evaporation in glacial summers due to cooler temperatures could have increased net infiltration (e.g., Koutsodendris et al., 2019). For example, pollen-based precipitation reconstructions from southern Iberia (Camuera et al., 2022) suggest that glacial net winter precipitation amount (23–19 kyr BP) was ~30% lower than during the late Holocene (0–4.2 kyr BP), while glacial net summer precipitation may have been double that of the late Holocene. This implies that during glacial

periods there may have been a potentially higher contribution of summer precipitation to Batura cave system than during interglacial periods. However, in our precipitation-driven Sr/Ca record, we observed a value of ~ 0.10 in glacial times and ~ 0.04 in interglacial times. This suggests that our proxy was not biased by seasonality in glacial-interglacial cycles, because increased glacial net summer precipitation should instead have led to low Sr/Ca values. This agrees with pollen-based summer temperature reconstructions for Lake Ohrid, Macedonia, in which MIS 12 summer temperature is around 12-18°C, which is not much lower than the current 20°C (Koutsodendris et al., 2019). Hence, net precipitation could have been low enough during summer. Infiltration water in the Batura cave system would still be dominated by the characteristics of winter half-year rainwater. We therefore argue that the Sr/Ca data can be considered as a (winter half-year) precipitation proxy during MIS 12 to MIS 11c.

The good alignment between Batura $\delta^{13}\text{C}$ and Sr/Ca (Figure 1b) in both interglacial and glacial periods suggests that the Batura $\delta^{13}\text{C}$ response to climate is not biased by any increase in summer net precipitation either. Overall, we argue that, even if shifts in the annual distribution of net precipitation could have happened on glacial-interglacial timescales, they do not seem to have materialized (see Lake Ohrid temperature mentioned above), and thus our overall interpretation of $\delta^{13}\text{C}$ and Sr/Ca stands. We have added the related discussion in Methods (Lines 297-310).

R2.3. Given the importance of the age model for conclusions, could you demonstrate that the results are similar if you choose a different age model algorithm? A comparison between outputs from StalAge and COPRA or OxCal (or similar algorithms) would strengthen your argument.

Figure R2.3 displays a comparison between our original StalAge and new OxCal age models. The 95% confidence levels of both the StalAge and OxCal age model overlap well, indicating that our conclusions would not change with the selection of the age model. Because the results are identical within errors, we use StalAge in the manuscript. We have updated **Supplementary Figure 4** and the related discussion in Lines 82-84.

Figure R2.3. Comparison between StalAge (red) and OxCal (blue) age models for BA7-1. Dots represent ^{230}Th ages, and error bars indicate 2-sigma uncertainties. Solid and dashed lines denote the median model age and the 95% confidence limits of the age model, respectively.

R2.4. The stable isotopes measurements were made on two different instruments. What measures were taken to ensure there are no systematic offsets between these instruments?

We calibrated our data using international standards, which – in principle – ensures that there is no offset between machines and laboratories. In this study, we used two instruments for isotopic analyses of speleothem carbonates. One is a Micromass IsoPrime mass spectrometer equipped with the Multicarb automatic system at the National Taiwan Normal University. The standards included NBS-19 ($\delta^{13}\text{C} = 1.95\text{‰}$, $\delta^{18}\text{O} = -2.20\text{‰}$), IAEA-CO1 ($\delta^{13}\text{C} = 2.49\text{‰}$, $\delta^{18}\text{O} = -2.4\text{‰}$), and MAB (in-house standard, marble collected from Taiwan; $\delta^{13}\text{C} = 3.4\text{‰}$, $\delta^{18}\text{O} = -6.88\text{‰}$). The other one is a Finnigan MAT-253 mass spectrometer linked to an online carbonate preparation system (Kiel-IV) at the College of Geological Science, Fujian Normal University. Three standards were NBS-19 ($\delta^{13}\text{C} = 1.95\text{‰}$, $\delta^{18}\text{O} = -2.20\text{‰}$), IAEA-603 ($\delta^{13}\text{C} = 2.46\text{‰}$, $\delta^{18}\text{O} = -2.37\text{‰}$), and CAI-13 (in-house standard; $\delta^{13}\text{C} = -10.73\text{‰}$; $\delta^{18}\text{O} = -9.46\text{‰}$). We have updated this information in Methods, see Lines 246-253 in our revised manuscript.

R2.5. The manuscript doesn't meet the latest IUPAC guidelines for reporting stable isotope ratio measurements (see Coplen, T.B., 2011, Rapid Commun. Mass Spectrom., 25, 2538). Please revise the methods section to include: a) what standards were used in addition to NBS-19 and what were their

isotopic values, b) based on what standard was the reproducibility of the isotopic measurements established, c) italicise delta symbol and leave space between number and permil symbol.

Corrected.

References:

- Camuera, J. et al. (2022). Past 200 kyr hydroclimate variability in the western Mediterranean and its connection to the African Humid Periods. *Scientific Reports*, *12*(1), 1–13.
- Hu, H.-M. et al. (2022). Split westerlies over Europe in the early Little Ice Age. *Nature Communications*, *13*(1), 4898.
- Koutsodendris, A., Kousis, I., Peyron, O., Wagner, B., & Pross, J. (2019). The Marine Isotope Stage 12 pollen record from Lake Ohrid (SE Europe): Investigating short-term climate change under extreme glacial conditions. *Quaternary Science Reviews*, *221*, 105873.

REVIEWERS' COMMENTS

Reviewer #1 (Remarks to the Author):

The authors have addressed comments and questions that I raised in the revisions, I appreciate their thoughtful and thorough revisions and discussion, both in the manuscript and in the letter to reviewer document. I would recommend the manuscript for publication.

Reviewer #2 (Remarks to the Author):

I am satisfied with the revisions made by the authors in response to the comments of both reviewers. This is a high-quality manuscript which I recommend for publication.

Kind regards,
Vasile Ersek